# The self-reported driving and pedestrian behaviour of adults with developmental coordination disorder

**Isobel Shipley**[1], **Maaike Esselaar**[1], **Samuel Wood**[1], **Johnny V. V. Parr**[1], **David J. Wright**[2], **Greg Wood**[1] *

**1** Department of Sport and Exercise Sciences, Manchester Metropolitan University Institute of Sport, Manchester, United Kingdom, **2** Department of Psychology, Manchester Metropolitan University, Manchester, United Kingdom

* greg.wood@mmu.ac.uk

**Data Availability Statement:** All data are available via the Open Science Framework (See here: https://osf.io/7cb9w/).

## Abstract

### Background

Developmental coordination disorder (DCD) affects movement coordination, but little is known about how the condition impacts the behaviours of car drivers and pedestrians.

### Aims

This study examined the self-reported driving and pedestrian behaviours of adults with Developmental Coordination Disorder (DCD).

### Methods and procedures

One hundred and twenty-eight participants (62 adults with DCD vs. 66 TD adults) responded to an online survey asking them about their perceptions of confidence and self-reported driving and pedestrian behaviours in the real-world.

### Outcomes and results

Results suggested that adults with DCD felt less confident and reported more lapses in attention (e.g., forgetting where their car was parked) and errors (e.g., failing to check their mirrors prior to a manoeuvre) when driving compared to typically developed (TD) adults. Adults with DCD also reported feeling less confident and reported less adherence to road traffic laws (e.g., not waiting for a green crossing signal before crossing the road) when walking as pedestrians.

### Conclusions and implications

These results offer some much-needed insight into the behaviours of those with DCD outside of the laboratory environment and underline the need for research investigating the driving and pedestrian behaviours of individuals with DCD in 'real-world' contexts.

**Funding:** The author(s) received no specific funding for this work.

**Competing interests:** The authors have declared that no competing interests exist.

## Introduction

Developmental Coordination Disorder (DCD), also known as dyspraxia, is a neurodevelopmental disorder that is primarily characterised by poor motor control and coordination. Those with DCD experience movement difficulties that are substantially below that expected given the individual's chronological age and which occur despite typical levels of neurological and intellectual function [1]. As well as impaired motor control, individuals with DCD often have poor spatial awareness [2], a reduced ability to judge distances [3], poor reaction time [4] and a reduced ability to process multiple stimuli [5]. It is perhaps unsurprising then that individuals with DCD often struggle to learn and perform activities of daily living, struggle when playing sports and struggle when navigating complex real-world environments [1].

One complex, real-world environment that has received research attention in relation to DCD is driving a car. Experimental studies using driving simulators have previously shown that drivers with DCD have more collisions, drive more slowly, and drive closer to on-coming traffic [6]. Drivers with DCD also show larger variance in car heading and make more steering adjustments on straight roads [7], show difficulties in steering when turning bends and take 50% more time to react to pedestrians in their path [8], compared to drivers without DCD. To date, only one study has attempted to document the driving behaviour of individuals with DCD in real-world driving contexts outside of the laboratory. Kirby, Sugden, and Edwards [9] used a semi-structured questionnaire to examine the driving experiences of young adults (17–25 years) with DCD. They found that drivers with DCD had felt less confident and needed a greater number of attempts to pass their theory and practical tests, drove significantly fewer miles per week, and experienced *fewer* traffic collisions and parking violations compared to typically developed adults. Although there is some disparity in these research findings, it is clear that driving is very challenging for individuals with DCD. Additionally, very little is known about *why* individuals with DCD drive differently (i.e., faster, or slower) or have more or less traffic violations compared to 'typically developed' (TD) adult drivers. These driving behaviours might be attributable to differences in confidence, aggressive driving behaviours, lapses in attention, or simply that individuals with DCD make more driving errors.

There is also a paucity of research that has explored the pedestrian behaviour of adults with DCD in the real-world. Pedestrians are individuals who walk or travel on foot, and their behaviour is influenced by a variety of factors, including cultural norms, environmental conditions, personal motivations, and situational contexts. While there is extensive evidence to suggest that children [e.g., 10,11] and adults [e.g., 12,13] with DCD exhibit different gait characteristics when walking in laboratory environments, little is currently known about how these differences in walking translate to pedestrian behaviour in complex real-world environments that require avoiding obstacles or people, avoiding distraction, and judging speed and distance when avoiding traffic. Previous work in relation to pedestrian behaviours of people with DCD has mainly focused on the road crossing ability of children. Findings from experimental research has suggested that children with DCD leave considerably longer temporal crossing gaps to cross than TD children [14]. It has been shown that children with DCD perceived road crossing as more dangerous, walk more slowly and leave significantly *less* time to cross the road safely [3]. In the only study to examine road crossing in the real-world, Wilmut and Purcell [15] used questionnaires to survey adults with DCD and parents of children with DCD about their road crossing behaviours. Findings showed that adults with DCD thought that the condition impaired their road crossing ability, reported low confidence in their road crossing ability, and exhibited a greater propensity to engage in dangerous crossing behaviours (i.e., forgetting to look, running without looking, crossing between cars, crossing when they can't see) despite understanding the risky nature of these behaviours and the likelihood of

accidents. While road crossing is an inherently dangerous part of being a pedestrian, there are other aspects of pedestrian behaviour that have yet to be assessed in the adult DCD population. For example, little is known about how confident individuals feel as pedestrians generally (i.e., outside of road crossing ability), their propensity to conform to norms of pedestrian behaviour (e.g., waiting for the appropriate signal to cross), to engage in risky pedestrian behaviours like wearing headphones or looking at their phone when walking, or even to exhibit aggressive pedestrian behaviours. Finally, the investigation conducted by Wilmut and Purcell [15] did not include a control group and it is therefore difficult to assess the self-reported pedestrian behaviours reported compared to TD adult pedestrians.

The aim of this study was therefore to explore the self-reported driving and pedestrian behaviours of adults with DCD to gain an understanding of their perception of the way in which DCD impacts how they navigate these real-world environments. Due to the significant challenges in carrying out this work in real-world driving and pedestrian contexts, capturing the lived experience of individuals with DCD through self-report will further inform the limited research in this area and help to facilitate future more ecologically valid studies in real-world-contexts. Based on previous research, it was hypothesised that adults with DCD would report lower confidence when driving and report a different number of traffic collisions compared to TD adult drivers. We further hypothesised that these increased collisions would be underpinned by more lapses in attention, more driving errors, and more driving violations. In relation to pedestrian behaviour, based on previous research we hypothesised that adults with DCD would report lower confidence as pedestrians, would suffer more lapses in attention (e.g., forgetting to look), engage in more risky behaviours (e.g., wearing headphones while walking) and exhibit less positive pedestrian behaviour (e.g., less adherence to pedestrian traffic laws) compared to TD adult pedestrians.

## Methods

### Participants

One hundred and twenty-eight adults with or without DCD were recruited via an online survey that was distributed by word of mouth, university email addresses, and social media platforms. Convenience sampling was used as it offers advantages when studying niche populations or groups that are small or challenging to access, like adults with DCD, which often makes probability sampling impractical. This sampling method has also been used in similar studies on driving and pedestrian behaviour in DCD populations [9,15]. Participants with DCD were required to confirm they had previously received a formal DCD diagnosis. Recruitment took place between December 2022 and March 2023. The demographics of each group are presented in Table 1. Participants who held a driver's licence answered items related

**Table 1. Participant demographics.**

|  | DCD Drivers | TD Drivers | DCD Pedestrians | TD Pedestrians |
|---|---|---|---|---|
| **Number of participants** | 46 | 58 | 62 | 66 |
| **Mean age (yrs.)** **(SD)** | 41 (18) | 41 (13) | 38 (14) | 40 (18) |
| **Number of years held a driver licence.** **(yrs.)** **(SD)** | 17 (13) | 19 (17) | - | - |
| **Gender** | Male = 9 Female = 36 Other = 1 | Male = 26 Female = 32 Other = 0 | Male = 14 Female = 46 Other = 2 | Male = 28 Female = 38 Other = 0 |

to being a driver and a pedestrian, those who did not hold a licence answered items only responded to questions related to being a pedestrian. Ethics was approved by the institutional ethics committee at Manchester Metropolitan University (ID:48614). Written informed consent was taken from all participants prior to data collection.

## Procedure

The survey was presented to participants using the online platform https://www.jisc.ac.uk/. The questionnaire used can be found in Supplementary Material.

## Measures

**Confidence and driving collisions.** Confidence when driving was measured by asking participants "How confident do you feel when driving?" and participants responded on a 5-point Likert scale from 1 (Not at all) to 5 (Extremely confident). The number of collisions was measured by asking "How many collisions with other cars, people or objects have you experienced when you have been driving?". Confidence as a pedestrian was measured by asking "How confident do you feel when crossing the road or walking next to the road?" and participants responded on a 5-point Likert scale from 1 (Not at all) to 5 (Extremely confident).

**Driving behaviour.** The Driver Behaviour Questionnaire (DBQ) [16] was used to assess self-reported driving behaviour. The DBQ contained 24 items related to four different behaviours: errors, lapses of attention, ordinary violations, and aggressive violations. Participants answered on a Likert scale (0 = never to 5 = always) and the mean score for each driving behaviour was calculated by aggregating responses from questions related to each behaviour [17]. Definitions and example questions for each behaviour are shown in Table 2. A meta-analysis of 174 studies revealed the DBQ to have good predictive validity of road traffic accidents [18].

**Pedestrian behaviour.** The Pedestrian Behaviour Questionnaire (PBQ) [19] was used to assess self-reported pedestrian behaviour. All participants answered 21 items from PBQ which contains items on four different behaviours: positive behaviours, violations, lapses of attention, and aggressive behaviours. The mean score for each pedestrian behaviour was calculated by aggregating responses from questions related to each behaviour [19]. Definitions and example questions for each behaviour are shown in Table 3. Participants answered on a Likert scale (0 = never to 5 = always). The PBQ has been shown to have good construct validity and test-retest reliability [19].

## Data analysis

Internal consistency and scale reliability was determined for each subscale of each questionnaire using Cronbach's alpha. For the DBQ, errors (α = .76), lapses of attention (α = .80),

**Table 2. Definition of the types of driving behaviour with example questions from the DBQ.**

| Driving Behaviour | Definition | Example |
|---|---|---|
| Errors | Deficiency in knowledge of traffic rules and/or in the inferential processes involved in making a decision. | 'Fail to check the rear-view mirror before a manoeuvre' |
| Lapses of attention | Unintentional deviation from practices related to a lack of concentration on the task, forgetfulness. | 'Forget where you left your car in the car park' |
| Ordinary violations | Deliberate deviation from social rules without intention to cause injury or cause damage. | 'Overtake a slow driver on the inside' |
| Aggressive behaviour | A tendency to misinterpret other road users' behaviour resulting in the intention to annoy or danger. | Race away from the traffic lights to beat another driver' |

**Table 3. Definition of the types of pedestrian behaviour with example questions from the PBQ.**

| Pedestrian Behaviour | Definition | Example |
|---|---|---|
| Positive behaviour | Behaviour that seeks to avoid violation or error and/or seeks to ensure traffic rules compliance | I cross the street after all the vehicles are stopped and the pedestrian light is green. |
| Violations | Deliberate deviation from social rules without intention to cause injury or cause damage. | I cross the street, talking on a cell phone or listening to music with my headphones. |
| Lapses of attention | Unintentional deviation from practices related to a lack of concentration on the task; forgetfulness | I follow other people who cross the street unsafely in dangerous situations. |
| Aggressive behaviour | A tendency to misinterpret other road users' behaviour resulting in the intention to annoy or endanger others. | I get angry with other road users (driver, pedestrian, cyclist) and insult them. |

ordinary violations ($\alpha$ = .84) and aggressive behaviour ($\alpha$ = .81), were found to be reliable and consistent. For the PBQ, scales related to positive behaviour ($\alpha$ = .70), violations ($\alpha$ = .77), lapses of attention ($\alpha$ = .81), and aggressive behaviour ($\alpha$ = .71) were found to be reliable and consistent. These values represent acceptable-to-good internal consistency across all sub-scales [20].

As data were non-parametric, separate Mann-Whitney U tests were run to determine the differences between the DCD and TD groups for confidence, number of collisions, and for each subscale of the DPQ and the DBQ.

## Results

### Driving behaviour

Adults with DCD self-reported less confidence when driving ($z$ = 3.35, $p$ < .001) compared to TD adults. There was no significant difference between the number of driving collisions reported between DCD and TD individuals ($z$ = 1.23, $p$ = .220).

Analysis of the DBQ revealed that there was no significant difference between driving violations ($z$ = 1.40, $p$ = .161) or aggressive driving behaviours ($z$ = 0.65, $p$ = .514) between DCD and TD groups. However, adults with DCD reported experiencing more self-reported driving errors ($z$ = 2.01, $p$ = .045) and more lapses in attention ($z$ = 5.69, $p$ < .001) compared to TD adults (Table 4).

**Table 4. Self-reported confidence when driving, number of collisions and driver behaviours for DCD and TD groups (\*$p$ < .05, \*\* $p$ < .001).**

| | DCD Drivers | TD Drivers |
|---|---|---|
| Confidence when driving | 2.98** (1.13) | 3.97 (1.19) |
| Number of collisions | 1.67 (2.49) | 1.03 (1.20) |
| Errors | 1.94* (0.15) | 1.63 (0.21) |
| Lapses of attention | 2.87** (0.11) | 1.89 (0.16) |
| Ordinary violations | 1.76 (0.25) | 1.93 (0.13) |
| Aggressive behaviours | 1.84 (0.13) | 1.92 (0.16) |

## Pedestrian behaviour

Adults with DCD self-reported less confidence ($z = 4.08$, $p < .001$) and reported significantly less positive pedestrian behaviours as measured by the PBQ ($z = 2.33$, $p = .020$) compared to their TD counterparts. Further analyses revealed no significant differences were evident in violations (z = 1.19, $p = .232$), lapses in attention ($z = 1.26$, $p = .206$) or aggressive behaviours ($z = 1.28$, $p = .201$) between groups.

## Discussion

The aim of this study was to understand more about the self-reported driving and pedestrian behaviours of adults with DCD. Regarding self-reported driving behaviours, it was hypothesised that adults with DCD would report lower confidence when driving and experience a different number of collisions. Results provided some support for these hypotheses in that adults with DCD reported lower confidence when driving compared to TD adults. There were, however, no differences in the number of self-reported collisions between the two groups, yet adults with DCD did report experiencing significantly more lapses of attention and making more errors when driving (see Table 3).

Lower confidence in individuals with DCD is well documented both as a general self-concept [21] and a domain-specific construct related to driving [9]. Lower confidence in adult drivers with DCD is intuitively understandable given their previous experiences of needing a greater number of attempts to pass their theory and practical tests [9], evidence that suggests they may experience more traffic collisions [6] and their general experience of struggling with motor tasks as a result of having lived with DCD all their lives [1]. This lower confidence may also be a reason why adults with DCD self-report driving slower and less often than typically developing adult drivers [9]. The findings from the DBQ indicate that adult drivers with DCD self-report experiencing more lapses of attention and making more errors when driving. Given their awareness of these issues, it is understandable that they are less confident when driving. It appears, therefore, that adult drivers with DCD may develop compensatory strategies to mitigate these issues, such as only driving when necessary and lowering their general driving speed when they do drive. Use of such compensatory strategies by adult drivers with DCD may explain why our findings show no differences in self-reported traffic collisions between the two groups. Although speculative, children and adults with DCD are known to develop compensatory strategies to cope with and overcome their motor coordination difficulties [22]) and so it is plausible that this may extend to driving behaviours.

In relation to pedestrian behaviour, it was hypothesised that adults with DCD would report less confidence as pedestrians and exhibit more risky behaviours compared to TD adults. Support was found for these predictions as pedestrians with DCD did indeed report lower levels of confidence and reported exhibiting less positive pedestrian behaviours compared to TD adults (Table 5). This lower confidence suggests that these self-perceptions are not just related to driving but extend to pedestrian behaviours too. This again corroborates previous research reporting lower levels of general self-confidence in individuals with DCD [21] and this may be related to the negative experiences individuals with DCD have perhaps faced as children and adults during road crossing [15,23]. Adults with DCD also reported engaging in significantly fewer positive (i.e., riskier) pedestrian behaviours compared to TD adults. The behaviours centre around observing the priorities of traffic, taking greater care at night-time, crossing in appropriate places and at appropriate times (e.g., when faced with a green walk signal), averting dangerous crossing situations, and attempting to estimate vehicle speeds and crossing gaps. Interestingly, many of these 'risky' behaviours have also been shown in previous experimental research [14,15,23,24]. Previous research has suggested that adults with DCD

**Table 5. Self-reported confidence as a pedestrian and pedestrian behaviours for DCD and TD groups (*$p < .05$, ** $p < .001$).**

|  | DCD Pedestrians | TD Pedestrians |
|---|---|---|
| Confidence when walking | 3.05** <br>(1.11) | 4.26 <br>(1.00) |
| Positive behaviours | 3.09* <br>(0.62) | 3.40 <br>(0.77) |
| Violations | 2.29 <br>(0.83) | 2.12 <br>(0.85) |
| Lapses of attention | 1.66 <br>(0.93) | 1.76 <br>(0.81) |
| Aggressive behaviours | 1.58 <br>(0.83) | 1.48 <br>(0.84) |

understand the risky nature of these behaviours and the likelihood of accidents [14]; our results also show that they are aware that they engage in these risky behaviours when pedestrians. As the data from the PBQ indicates no differences in lapses of attention, violations, or aggressive behaviours between adults with and without DCD (Table 5), it is likely that other factors may underpin this riskier pedestrian behaviour reported by adults with DCD. One possibility is that although adults with DCD are aware of what constitutes safe pedestrian behaviour, it is only in practice, where the execution of these behaviours is dependent on their impaired perceptual abilities, that they find themselves in compromised and potentially unsafe pedestrian scenarios. In essence, it may not be that they wilfully engage in unsafe pedestrian behaviours, but rather that they often find themselves in these situations due to their impaired perceptual abilities.

Taken together these findings highlight a general feeling of lower confidence when navigating real-world environments in adults with DCD. This is in sharp contrast to previous experimental studies using simulated (i.e., computer-based) pedestrian [24] and driving environments [6–8] that found no differences in confidence between individuals with and without DCD. An explanation for these contradictory findings could therefore be related to differences in study methodologies. For example, laboratory-based experiments in virtual environments have no real physical consequences of the driving or pedestrian behaviours performed by participants. This may inflate the confidence of individuals with DCD and is maybe the cause of reckless behaviours witnessed in both simulated driving [6–8] and simulated pedestrian contexts [24]. There are obvious limitations to self-report measures used in this study also, and the disparity between self-reported and objective measures of driving behaviour and road safety have been documented [25]. Therefore, there is a clear need for the examination of the behaviour of individuals with DCD in everyday, real-world environments. While such research offers some unique challenges, the availability and accessibility of technologies (e.g., mobile eye-tracking) may offer solutions for understanding the perceptual-cognitive strategies used by adults with DCD in real-word driving and pedestrian contexts. Future work should also investigate the relationship between anxiety and confidence in individuals with DCD when navigating the real world. Anxiety typically shares an inverse relationship with perceptions of self-confidence [26], so it is possible that lower confidence in driving and pedestrian context could be anxiety-induced. Given the reported past experiences in these contexts (e.g., more collisions) this interaction is likely and worthy of future investigation. Finally, a limitation of this work is that we did not account for co-occurring conditions like ADHD that are prevalent in the DCD population and have been previously shown to affect at least some aspects of road crossing behaviour (i.e., looking behaviours [15]). Future research should take

greater consideration of co-occurring conditions and their mediating role in pedestrian and driving behaviour.

## Author Contributions

**Conceptualization:** Isobel Shipley, Maaike Esselaar, Samuel Wood, Johnny V. V. Parr, David J. Wright, Greg Wood.

**Data curation:** Isobel Shipley, Johnny V. V. Parr, David J. Wright, Greg Wood.

**Formal analysis:** Greg Wood.

**Investigation:** Isobel Shipley, Johnny V. V. Parr, Greg Wood.

**Methodology:** Maaike Esselaar, Samuel Wood, Johnny V. V. Parr, David J. Wright, Greg Wood.

**Project administration:** Isobel Shipley.

**Resources:** Isobel Shipley.

**Supervision:** Samuel Wood, Greg Wood.

**Writing – original draft:** Isobel Shipley, Greg Wood.

**Writing – review & editing:** Isobel Shipley, Maaike Esselaar, Samuel Wood, Johnny V. V. Parr, David J. Wright, Greg Wood.

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
