## [Decision Letter · Decision Letter 0]

22 Nov 2023

PONE-D-23-31342The self-reported driving and pedestrian behaviour of adults with Developmental Coordination DisorderPLOS ONE

Dear Dr. Wood,

Thank you for submitting your manuscript to PLOS ONE. After careful consideration, we feel that it has merit but does not fully meet PLOS ONE’s publication criteria as it currently stands. Therefore, we invite you to submit a revised version of the manuscript that addresses the points raised during the review process.

**ACADEMIC EDITOR: **1. Clarify Study Objective: The objective of the study needs to be clearly defined. Understanding self-reported behavior should be framed within a broader, clearly articulated objective (required for acceptance).

2. Methodology and Data Collection: Address concerns about the appropriateness of the methodology, especially regarding the reliance on self-reported behaviors. Clarify the sampling method used and its adequacy for the study's goals (required for acceptance).

3. Representation of Previous Studies: Accurately and clearly represent previous studies in the field, highlighting the novel aspects of the current study (required for acceptance).

4. Consistency and Reliability of DBQ and PBQ: Ensure and demonstrate the consistency and reliability of responses in the Driver Behaviour Questionnaire (DBQ) and Pedestrian Behaviour Questionnaire (PBQ) (required for acceptance).

5. Gender Distribution Control: Explain how the differences in gender distribution between DCD and non-DCD samples were accounted for in the study (required for acceptance).

6. Detailed Analysis of DCD Group: Provide more detailed information about the group with Developmental Coordination Disorder (DCD), including inclusion criteria and any co-occurring conditions (recommended modifications).

7. Recruitment Methods for Both Groups: Elaborate on how the DCD group and the control group were recruited, ensuring transparency in the process (recommended modifications).

8. Influence of Age and Driving Experience: Investigate and discuss how age or years of driving experience might influence confidence or behavior, especially in the DCD group (recommended modifications).

9. Limitations of Self-Report in Study Design: Include a discussion on the limitations of self-reporting, particularly in the context of comparing lab-based and real-world situations (recommended modifications).

10. Aggregation of Questions within Constructs: Clarify how questions within the same construct were aggregated and the impact of this approach on statistical power (recommended modifications). 

We look forward to receiving your revised manuscript.

Kind regards,

Marcus Tolentino Silva, Ph.D.

Academic Editor

PLOS ONE

Reviewers' comments:

Reviewer's Responses to Questions

**Comments to the Author**

1. Is the manuscript technically sound, and do the data support the conclusions?

Reviewer #1: Yes

Reviewer #2: No

2. Has the statistical analysis been performed appropriately and rigorously? 

Reviewer #1: Yes

Reviewer #2: No

3. Have the authors made all data underlying the findings in their manuscript fully available?

Reviewer #1: Yes

Reviewer #2: Yes

4. Is the manuscript presented in an intelligible fashion and written in standard English?

Reviewer #1: Yes

Reviewer #2: Yes

5. Review Comments to the Author

Reviewer #1: This is an interesting paper which addresses an area of research which is understudied. The manuscript is well written and clear in the methodologies used, however, it is lacking in detail in places and I think a more detailed analysis of the data would really support the conclusions. There are also some other points that I would like the authors to consider before I would be happy to accept this paper for publication:

1. I don't believe the authors have well represented some of the previous studies within this field. For example, the Wilmut and Purcell study cited did consider confidence and also adherence to road regulations (waiting for the green man etc) - what this paper adds that the previous paper did no consider was an inclusion of a control group. These nuanced differences should be really clear so that there is a clear and accurate novelty of the study. This is just one example, a discussion of the relevant papers in more depth would really support the argument for the study.

2. I would like to know more about both groups. Firstly the group with DCD - what information was given with regards to this, did these individuals have to have a diagnosis of DCD or just suspect that they had DCD? Where any co-occurrences present? This is particularly important as previous work have considered the co-occurrence of ADHD and ASD in road crossing behaviour and so if this study did not consider those co-occurrences some mention of this should be provided in the discussion.

3. I would also be interested to know how the two groups were recruited - through the same means? Or were the control group individuals known to the researchers - very little information is given about this.

4. I wonder whether age or years driving influences confidence or behaviour? This is a statistical consideration that the authors have not included - but I would imagine confidence, especially in driving might increase with age in controls - what is the relationship in the DCD group.

5. I was really pleased to see that the authors made comparisons between lab based (no risk) and real world situations. However, no discussion is included regarding the limitations of self-report in this context.

Reviewer #2: This study investigated the differences between the behaviours of drivers and pedestrians with and without Developmental Coordination Disorder (DCD). While the study subject is very interesting, there are major flaws with the study:

- The objective of the study is not clear. Understanding self-reported behaviour of a group of drivers and pedestrians is not an objective, but a task. What is it that the authors aim to understand (fundamentally) using these self-reported behaviours?

- The methodology is not appropriate to address the research gap. As I understand from the manuscript, the identified research gap is that most studies have investigated the behaviours of DCD drivers and pedestrians in simulated environments, which may not be exactly the same as in real-world environments. However and as the authors have mentioned themselves, studying self-reported behaviours has biases and lack of reliability as well. How can the authors claim that studying self-reported behaviours is addressing the gap above?

- Reliance on self-reported behaviours is even more acute when it comes to the DCD people. As the authors have mentioned, DCD is a neurodevelopmental disorder which, in addition to impaired motor control, may impact cognitive functionality (e.g. spatial awareness, ability to judge distances, reaction time, ability to process multiple stimuli) too. This essentially creates a chicken-egg problem. The self-reported behaviours may well have been influenced by these cognitive disabilities.

- What sampling method was used to collect data? From what I can see, the authors have used convenience sampling, which is characterized with insufficient power to identify differences between population subgroups.

- The gender distribution is very different among the DCD and non-DCD samples. How did the authors control for that?

- The DBQ and PBQ have many items within the same construct. I cannot see the consistency and reliability tests for these items. Did the authors ensure consistency and reliability in the responses?

- The DBQ and PBQ have many items within the same construct. I can see that the authors have applied the Mann-Whitney tests on the whole construct (e.g. errors, lapses, etc.) but how did they aggregate the questions within he same construct? Did they take the average (which by and of itself may have resulted in lack of statistical power).

6. PLOS authors have the option to publish the peer review history of their article (what does this mean?). If published, this will include your full peer review and any attached files.

Reviewer #1: No

Reviewer #2: No

---

## [Decision Letter · Decision Letter 1]

22 Feb 2024

PONE-D-23-31342R1The self-reported driving and pedestrian behaviour of adults with Developmental Coordination DisorderPLOS ONE

Dear Dr. Wood,

Thank you for submitting your manuscript to PLOS ONE. After careful consideration, we feel that it has merit but does not fully meet PLOS ONE’s publication criteria as it currently stands. Therefore, we invite you to submit a revised version of the manuscript that addresses the points raised during the review process.

We look forward to receiving your revised manuscript.

Kind regards,

Marcus Tolentino Silva, Ph.D.

Academic Editor

PLOS ONE

Journal Requirements:

**Additional Editor Comments:**

1. Explicitly articulate the rationale behind the significance and contribution of your work in the manuscript.

2. Ensure this discussion directly responds to the reviewer's comment, highlighting the novelty and utility of your findings.

3. Include a clear justification and discussion of your use of convenience sampling within the manuscript.

4. Discuss the implications of this sampling method on the validity and generalizability of your results.

5. If applicable, describe any strategies employed to mitigate potential biases associated with convenience sampling.

6. Clarify how questions within the same construct were aggregated.

7. Provide justification for this methodological choice, if not already done, to enhance the reader's understanding of your analytical approach.

Reviewers' comments:

Reviewer's Responses to Questions

**Comments to the Author**

1. If the authors have adequately addressed your comments raised in a previous round of review and you feel that this manuscript is now acceptable for publication, you may indicate that here to bypass the “Comments to the Author” section, enter your conflict of interest statement in the “Confidential to Editor” section, and submit your "Accept" recommendation.

Reviewer #1: All comments have been addressed

Reviewer #2: (No Response)

2. Is the manuscript technically sound, and do the data support the conclusions?

Reviewer #1: Yes

Reviewer #2: Partly

3. Has the statistical analysis been performed appropriately and rigorously? 

Reviewer #1: Yes

Reviewer #2: N/A

4. Have the authors made all data underlying the findings in their manuscript fully available?

Reviewer #1: Yes

Reviewer #2: Yes

5. Is the manuscript presented in an intelligible fashion and written in standard English?

Reviewer #1: Yes

Reviewer #2: Yes

6. Review Comments to the Author

Reviewer #1: I would like to thank the authors for addressing my comments, I am happy to recommend this for publication

Reviewer #2: Thank you for addressing my comments. However, in addition to the response letter, the authors must bring their responses in the manuscript as well:

• I would like to see the authors reasoning on the value of their work (i.e. their response to my second comment) in the manuscript as well. I think this is a key discussion that must be in the manuscript to pinpoint the contribution of this work.

• Likewise, the authors must bring the discussion around convenience sampling inside the manuscript too.

• The way they have aggregated the questions within the same construct (taking the average) must be stated in the manuscript too.

7. PLOS authors have the option to publish the peer review history of their article (what does this mean?). If published, this will include your full peer review and any attached files.

Reviewer #1: No

Reviewer #2: No

---

## [Decision Letter · Decision Letter 2]

12 Mar 2024

The self-reported driving and pedestrian behaviour of adults with Developmental Coordination Disorder

PONE-D-23-31342R2

Dear Dr. Wood,

We’re pleased to inform you that your manuscript has been judged scientifically suitable for publication and will be formally accepted for publication once it meets all outstanding technical requirements.

Kind regards,

Marcus Tolentino Silva, Ph.D.

Academic Editor

PLOS ONE

Additional Editor Comments (optional):

Reviewers' comments:

Reviewer's Responses to Questions

**Comments to the Author**

1. If the authors have adequately addressed your comments raised in a previous round of review and you feel that this manuscript is now acceptable for publication, you may indicate that here to bypass the “Comments to the Author” section, enter your conflict of interest statement in the “Confidential to Editor” section, and submit your "Accept" recommendation.

Reviewer #1: All comments have been addressed

Reviewer #2: All comments have been addressed

2. Is the manuscript technically sound, and do the data support the conclusions?

Reviewer #1: Yes

Reviewer #2: Yes

3. Has the statistical analysis been performed appropriately and rigorously? 

Reviewer #1: Yes

Reviewer #2: Yes

4. Have the authors made all data underlying the findings in their manuscript fully available?

Reviewer #1: Yes

Reviewer #2: No

5. Is the manuscript presented in an intelligible fashion and written in standard English?

Reviewer #1: Yes

Reviewer #2: Yes

6. Review Comments to the Author

Reviewer #1: I didn't raise any issues in this second round and so I have nothing to comment on - I am still happy to recommend this paper for publication

Reviewer #2: I have no further comments, thank you for your efforts to address my comments.

7. PLOS authors have the option to publish the peer review history of their article (what does this mean?). If published, this will include your full peer review and any attached files.

Reviewer #1: No

Reviewer #2: No

---

## [Editor Report · Acceptance letter]

29 Apr 2024

PONE-D-23-31342R2 

PLOS ONE

Dear Dr. Wood, 

I'm pleased to inform you that your manuscript has been deemed suitable for publication in PLOS ONE. Congratulations! Your manuscript is now being handed over to our production team.

Kind regards, 

on behalf of

Prof Marcus Tolentino Silva 

Academic Editor

PLOS ONE